# A retrospective cohort study evaluating correlates of deep tissue infections among patients enrolled in opioid agonist treatment using administrative data in Ontario, Canada

Kristen A. Morin[1,2], Chad R. Prevost[1], Joseph K. Eibl[1], Michael T. Franklyn[1], Alexander R. Moise[1], David C. Marsh [1,3]*

1 Northern Ontario School of Medicine, Sudbury, ON, Canada, 2 Laurentian University, Sudbury, ON, Canada, 3 Canadian Addiction Treatment Centres, Richmond Hill, ON, Canada

* dmarsh@nosm.ca

## Abstract

### Objective

The objective of this study was to evaluate the relationship between individual characteristics and deep tissue infections in patients enrolled in opioid agonist treatment in Ontario, Canada.

### Methods

A retrospective cohort study was conducted on patients in opioid agonist treatment between January 1, 2011, and December 31, 2015 in Ontario, Canada. Patients were identified using data from the Ontario Health Insurance Plan Database, and the Ontario Drug Benefit Plan Database. We identified other study variables including all-cause mortality using data from the Registered Persons Database. Encrypted patient identifiers were used to link across databases. Logistic regression models were used to measure potential correlates of deep tissue infections.

### Results

An increase in the incidence of deep tissue infections was observed between 2011 and 2016 for patients on opioid agonist treatment. Additionally, age, sex, positive HIV diagnosis, and all-cause mortality was correlated with deep tissue infection in our study population.

### Conclusion

The study indicates factors that are associated with deep tissue infections in the opioid use disorder population and can be used to identify opportunities to reduce the incidence of new infections.

**Data Availability Statement:** All data is available in paper.

**Funding:** This study was funded by a Clinical Innovation Grant from the Northern Ontario Academic Medicine Association. It was also supported by the Institute for Clinical Evaluative Sciences (ICES), a non-profit research institute sponsored by the Ontario Ministry of Health and Long Term Care. The design and conduct of the study; collection, management, analysis, and interpretation of the data; and preparation and review of the manuscript were conducted by the authors independently from the funding sources.

**Competing interests:** Dr. David Marsh maintains the following roles: Chief Medical Director at CATC (Canadian Addiction Treatment Center), opioid agonist therapy provider. Dr. Marsh has no ownership stake in the CATC as a stipendiary employee. We do not foresee any conflict of interest as data will be made freely available to the public and neither the CATC, nor the Universities, have the ability to prevent publication and dissemination of knowledge. The authors have no conflicts declared. This does not alter our adherence to PLOS ONE policies on sharing data and materials

## Introduction

Deep tissue infections (DIT) are pathogenic infections involving subcutaneous tissues that may proliferate to surrounding tissues and muscles [1]. This is particularly due to the introduction of bacteria into the venous circulation or subcutaneous tissue, which is common among those who inject drugs [2, 3]. Multiple studies have indicated increasing rates of infective endocarditis [4–7], osteomyelitis [3, 8–15], and septic arthritis associated with intravenous drug use.

Individuals with opioid use disorder (OUD) have an increased risk of acquiring deep tissue infections due to the high prevalence of subcutaneous injections [3, 8–20]. OUD rates have risen to epidemic proportions in many communities in Ontario. For instance, opioid-related mortalities in Ontario, Canada, have risen from 571 per 100,000 people in 2010 to 865 per 100,000 people in 2016 [21]. Opioid Agonist Treatment (OAT) is currently the standard of care and the intervention with the best evidence for long term patient safety, social wellness, and physical health benefits for the treatment of OUD [22]. In many studies, OAT rates have been used to measure the prevalence of OUD in the population [23–25].

Rosenhall et al. demonstrated that patients with deep tissue infections are associated with injection drug use have high rates of readmissions to hospital, re-occurring infections and even death [26]. Therefore, it is important to understand which patient factors are correlated with acquiring deep tissue infections. Moreover, in a recent systematic review, Larney et al. stated that there were significant gaps in research pertaining to injection related injuries and disease in low- and middle-income countries, as well as gaps in research on the potential risk and protective factors of such diseases [27]. This study will focus on determining the incidence of IE, OM, and septic arthritis among individuals in OAT in Ontario, Canada, and determine which individual factors are correlated with deep tissue infections. We hypothesize that the incidence of deep tissue infections has increased over the last five years with the worsening of the opioid crisis. We also hypothesize that individual patient characteristics such as older age, and being female [28, 29], may be associated with a higher risk of DTI [27]. As well, we predict that living in northern Ontario may be associated with a higher risk of acquiring deep tissue infections since the incidence rate for new HIV is higher in the North [30, 31].

## Methods

### Study overview

This retrospective cohort study of patients on OAT was conducted between January 1, 2011, and December 31, 2015. Every patient in the study was followed for one year or more. If patients started on OAT at the end of 2015, they were followed until December 31, 2016. The number of people in OAT is commonly used as a proxy measure for estimating the prevalence of OUD in a population since it is difficult to estimate the true prevalence of patients not seeking treatment [23, 24, 32–34]. The first episode of OAT was used to identify patients, meaning that there was no previous history of OAT (including methadone or buprenorphine/naloxone) in the year before to the first treatment episode. Studies have shown that multiple treatment attempts are associated with a higher likelihood of positive outcomes [35–37]. Therefore, we chose to only include first-time OAT patients to eliminate bias associated with cases involving multiple treatment attempts. This study was approved by the Research Ethics Board of Laurentian University in Sudbury. This study followed the Strengthening the Reporting of Observational Studies in Epidemiology (STROBE), and Reporting of studies Conducted using Observational Routinely collected health Data (RECORDS) guidelines [38, 39].

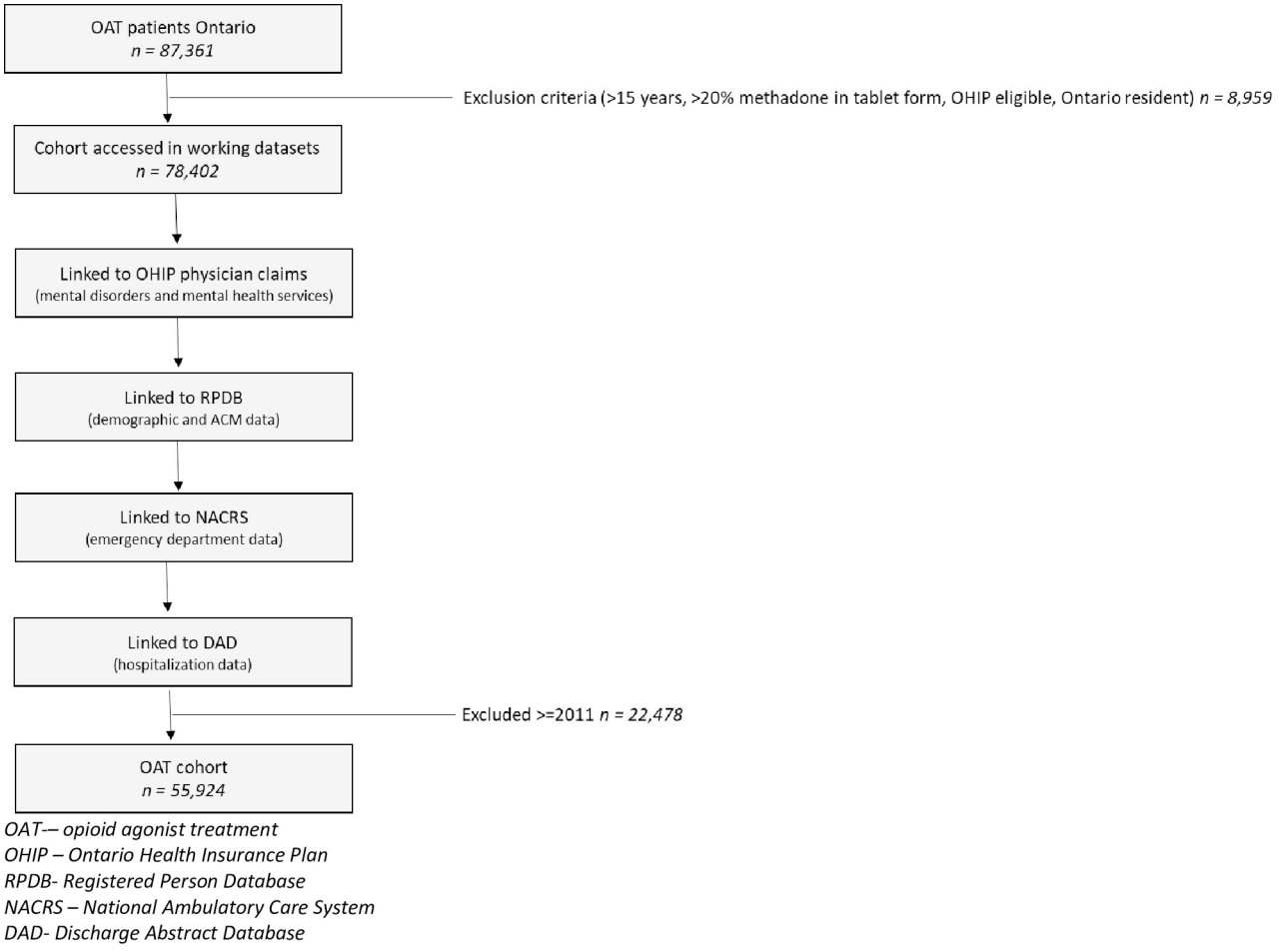

OAT-– opioid agonist treatment
OHIP – Ontario Health Insurance Plan
RPDB- Registered Person Database
NACRS – National Ambulatory Care System
DAD- Discharge Abstract Database

**Fig 1. Flow chart outlining data build including linkages.**

## Data sources

Individual-level administrative data from ICES was accessed remotely using a secure server. ICES, formally known as the Institute for Clinical and Evaluative Sciences, is an independent, non-profit research institute that collects and analyzes health care and demographic data for health system evaluation and improvement within the parameters of the Ontario Personal Health Information Privacy Act. ICES provided individual-level patient data collected from Ontario publicly funded health services. Patient information was linked anonymously across databases using encrypted 10-digit health card numbers. Linkages are presented in Fig 1.

All diagnostic information from physician visits were determined using billing data from the Ontario Health Insurance Plan (OHIP) database. Emergency department visits were identified using the Canadian Institute for Health Information National Ambulatory Care Reporting System (NACRS). Hospital admissions were identified using the Canadian Institute for Health Information Discharge Abstract Database. We obtained patients' location of residence and demographic information including all-cause mortality from the Ontario Registered Persons Database (RPDB), which contains unique data for each resident who has ever received insured health. Information on databases and study variables are available in S1 Appendix.

## Cohort definition

The cohort was created with the Ontario Drug Benefit Plan (ODB) database using drug identification numbers (DIN); and with the Ontario Health Insurance Plan (OHIP) database using physician billing codes including (Table 1): OAT monthly management codes (K682, K683, and K684), visit/consultation codes (A680 and A957) and, point of care testing codes (G040, G041, G042 or G043). A set of exclusion criteria was applied to define the main cohort. All patients under 15years of age were excluded (n = 2,535 patients). In Ontario, methadone used for addiction treatment is dispensed exclusively in liquid formulation. Therefore, patients with over 20% of their methadone dose in tablet formulation over a one year were excluded due to the likelihood that methadone was being administered for chronic pain (n = 5,560 patients). Additionally, patients who were not eligible for OHIP (n = 437 patients), non-Ontario residents (n = 427 patients) were excluded from the study. Patients identified from ODB (n = 1,383 patients), from OHIP (n = 30,124 patients) and patients who were identified in both databases (n = 24,417 patients) were combined to create the primary cohort (n = 55,924 patients).

## Correlates

**Age groups.** Previous research has shown a higher prevalence of OUD and accidental deaths related to opioids in persons aged 15–44 [40]; therefore, it was important to consider age group as a potential correlate for deep tissue infections. We used the RPDB database to create age groups. For relative measures of association, we used the 65 and over group as the reference category since the prevalence of OUD was the lowest in this group. All records contained information on patient age.

**Sex.** It was important to include sex as a potential correlate because studies have shown that sex and gender may influence outcomes for OAT patients [41, 42]. The sex variable was created from the RPDB database. For relative measures of association, we used males as the reference category. All records contained information on patient sex.

**Location of residence.** The location of residence variable was created using information from the RPDB. Postal codes were used to determined rurality and provincially defined health regions (Local Health Integration Networks) to determine the location [43]. At the time of the study Local Health Integration Networks (LHIN) are regional planning entities that plan and

**Table 1. Addiction medicine fee codes.**

| Addiction Medicine Fee Codes | |
|---|---|
| **OAT monthly management codes** | K682 Opioid Agonist Maintenance Program monthly management fee—intensive, per month |
| | K683 Opioid Agonist Maintenance Program monthly management fee—maintenance, per month |
| | K684 Opioid Agonist Maintenance Program—team premium, per month, to K682 or K683 |
| **OAT visit/consultation codes** | A680 Consultation |
| | A957 Repeat Consultation |
| **OAT point of care testing codes** | G040 Drugs of abuse screen, urine, must include testing for at least four drugs of abuse |
| | G041 Target drug testing, urine, qualitative or quantitative |
| | G042 Target drug testing, urine, qualitative or quantitative |
| | G043 Drugs of abuse screen, urine, must include testing for at least four drugs of abuse |

administer funding for health services across 14 defined geographic areas of Ontario. LHINs 13 and 14 (North East and North West LHIN) were used to define north Ontario, and the remainder (i.e., LHINs 1–12) of the LHINs were used to define southern Ontario. ICES uses the Statistics Canada Rural and Small Town definition to distinguish between rural and urban areas [44]. We created the location of residence variable from the two variables: north/south and rural/urban to form four categories used in the analysis: northern rural, northern urban, southern rural and southern urban. The location of residence was only considered at the index date (the time first seen for OAT treatment), and we did not account for the fact that the patient may have moved between rural and urban areas within the study period. For relative measures of association, we used the southern urban group as the reference category. The records with missing information on the location of residence were deleted (n = 3).

Northern, rural communities in Ontario have some of the highest rates of opioid-related deaths and physicians prescribing opioids in Ontario [32]; therefore, it was an important correlate to consider for this study.

**Income quintile.** Income, material deprivation, and socio-economic status have been linked to higher rates of OUD [45–47]. Since income can have a significant impact on health, it was included as a potential correlate. ICES uses neighborhood-level metrics (postal codes) from census data to create the income quintile variable. Therefore, it is important to note that information can be misclassified if a low income family was living in a neighborhood classified as being high income or vice versa. Income is classified into five categories: 1 (lowest), 2, 3, 4, 5 (highest). The income quintiles were chosen based on recent peer reviewed literature using ICES data [48–50], and the importance of placing this work in the context of other studies in the field. Income quintile was only considered at the index date, and we did not account for patients moving between neighborhoods. For relative measures of association, we used the highest income quintile as the reference category. It is known that low income and unemployment rates are disproportionately higher among OAT patients [51–53]. Therefore, the records with missing information on income quintile were re-assigned to the lowest income group (n = 687, 1.2%).

**Human Immunodeficiency Virus (HIV).** HIV can be an indication of the degree of morbidity of patients with OUD [54–56]; therefore, it was considered an important baseline covariate for this population. HIV status for patients in the cohort was derived from the OHIP database. ICES uses an algorithm that identifies an individual with HIV when there have been three physician claims in three years specific to HIV diagnosis [57]. HIV was a dichotomous baseline covariate for this study uploaded to the "master list" in the following categories: HIV yes or no. Since HIV status was used as a baseline covariate, we did not evaluate the incidence or impact of new HIV infections during the time course of the study. For relative measures of association, we used the group with a negative HIV status as the reference category.

**All-cause mortality.** Although studies have found that overdose is the most common cause of death among people who use opioids, [58, 59], previous studies have found that, other underlying causes of death, including circulatory and infections, are also common among people who use opioids [60–62]. Therefore, it was important to evaluate the correlation between deep tissue infections and all-cause mortality in our population. The all-cause mortality variable was calculated using data from the RPDB database by calculating the number of days to death date from the index for each patient in the cohort. If the patient had the event anytime between their index date and the end of the study period (December 31, 2016), we assigned a code as 1 (all-cause mortality) or 0 (no all-cause mortality). For relative measures of association, we used the group with no all-cause mortality as the reference category.

**Receiving continuous OAT.** Opioid use disorder is a chronic, relapsing condition [63], and patients often cycle through episodes of treatment and relapse [64]. Therefore, it was important to understand the association between being actively enrolled in OAT and deep

tissue infections in our cohort. Previous studies have demonstrated that OAT enrollment has been proven to decrease the frequency of injection drug use [65, 66] however, many patients continue to have IDU while in OAT. We identified an OAT treatment episode based on OHIP billing (from the OHIP database). All patients were followed after their first treatment episode, to a maximum follow-up date of December 31, 2016. The exact date of service was not available in the ICES datasets used for this study due to privacy reason. Rather, we were able to calculate an approximation of the time of a health service event based on a variable which had information relating to the number of days to service from the index data. We used the number of days to service date to query whether a patient was enrolled in OAT using 30 day intervals (i.e. if the days to service data was larger or equal to -30 and smaller or equal to 30 at the time that an OAT OHIP billing code appeared (see Table 1) then the patient was considered to be "actively enrolled in OAT for at least 30 days. If the days to service date was larger or equal to -30 and smaller or equal to 60 at the time that an OAT OHIP billing code appeared (see Table 1) then the patient was considered to be "actively enrolled in OAT for at least 60 days, etc. We then created a variable indicating whether a patient was actively engaged in OAT or not actively engaged in OAT.

## Outcome: Diagnosis of deep tissue infection

There were three outcomes analyzed for this study which were identified with the Ontario Health Insurance Plan database diagnostic codes: Infective endocarditis (OHIP fee code 429), Osteomyelitis (OHIP fee code 730), Septic Arthritis (OHIP fee code 711). A diagnostic code had to have appeared in any setting in the medical record for an individual patient to be considered as having any of the outcomes within one year of their index date. Infective endocarditis was included because it is a pathogenic infection of the endocardial tissue and heart valves. As the heart function becomes compromised, it can often lead to end-organ failure and life-threatening complications [67]. Osteomyelitis was included in the analysis because it is a pathogenic infection of the bone that is accompanied with inflammation and bone tissue necrosis [8, 68–70]. Septic Arthritis is a pathogenic infection of a joint and synovial fluid that may lead to rapid and progressive damage to the joint [8, 71]. The three infections were chosen because they are prevalent among people with opioid use disorder, and the prevalence of all three infections has been steadily increasing globally [4, 5, 16, 17, 71–73].

## Statistical analysis

All statistical analysis was conducted using SAS Version 9.4 [74]. The incidence of having a diagnosis of one or more deep tissue infections by year in our study population was calculated. Additionally, descriptive statistics were calculated for patients with one or more diagnoses of any deep tissue infections, patients with no diagnosis of any deep tissue infections, then for all three infections separately. Chi-square statistic was used to compare categorical variables. Logistic regression models were used to test the association between patient characteristics and each distinct deep tissue infection. Parameter estimates were used to calculate odds ratios (OR). We calculated 95% confidence intervals (CI) for each OR. Results were considered statistically significant, where the p value was less than 0.05.

## Results

### Patient characteristics

A total of 55,924 patients were enrolled in OAT at least once between 2011 and 2015. In total, 6,784 (12.1%) of patients had one or more deep tissue infections, 5,642 (10.09%) patients

**Table 2. Frequencies, descriptive statistics and Chi-square statistics of each patient characteristics who.**

| | One or more Deep Tissue Infections | No Deep Tissue Infection | P | Infective endocarditis | Osteomyelitis | Septic Arthritis | P |
|---|---|---|---|---|---|---|---|
| | n = 6,784 (12.1%) | n = 49,140 | | n = 5,642 | n = 1,236 | n = 576 | |
| **Age n (%)** | | | <0.01 | | | | <0.01 |
| 15 to 24 | 774 (11.4) | 9467 (19.3) | | 692 (12.3) | 95 (7.7) | 47 (8.2) | |
| 25 to 34 | 1367 (20.2) | 17640 (35.9) | | 1118 (19.8) | 243 (19.7) | 113 (19.6) | |
| 35 to 44 | 1422 (20.9) | 10673 (21.7) | | 1116 (19.8) | 307 (24.8) | 163 (28.3) | |
| 45 to 54 | 1758 (25.9) | 8043 (16.4) | | 1430 (25.3) | 351 (28.4) | 154 (26.7) | |
| 54 to 65 | 983 (14.5) | 2703 (5.5) | | 837 (14.8) | 189 (15.3) | 68 (11.8) | |
| 65+ | 480 (7.1) | 614 (12.5) | | 449 (8.0) | 51 (4.1) | 31 (5.4) | |
| **Sex n (%)** | | | | | | | 0.65 |
| Male | 4213 (62.1) | 32016 (65.2) | <0.01 | 3435 (60.9) | 799 (64.6) | 370 (64.2) | |
| Female | 2571 (37.9) | 17124 (34.8) | | 2207 (39.1) | 437 (35.4) | 206 (34.8) | |
| **Place of residence n (%)** | | | <0.01 | | | | 0.01 |
| Southern Urban | 5435 (12.6) | 37661 (87.4) | | 4555 (76.3) | 968 (2.3) | 443 (1.0) | |
| Southern Rural | 545 (11.7) | 4153 (8.7) | | 472 (80.1) | 78 (1.7) | 39 (0.8) | |
| Northern Urban | 542 (10.2) | 4752 (9.7) | | 411 (67.3) | 125 (2.4) | 75 (1.4) | |
| Northern Rural | 261 (9.2) | 2572 (5.2) | | 203 (70.7) | 65 (2.3) | 19 (0.7) | |
| **Income n (%)** | | | 0.25 | | | | 0.34 |
| 5 (highest) | 789 (11.6) | 5477 (9.3) | | 677 (12.0) | 118 (9.5) | 57 (9.9) | |
| 4 | 945 (13.9) | 7147 (14.5) | | 800 (14.2) | 170 (13.8) | 74 (12.8) | |
| 3 | 1172 (17.3) | 8800 (17.9) | | 996 (17.6) | 195 (15.8) | 96 (1.7) | |
| 2 | 140 (2.1) | 10959 (21.9) | | 1257 (22.2) | 281 (22.7) | 142 (24.7 | |
| 1 (lowest) | 2275 (33.5) | 16133 (32.8) | | 1862 (33.0) | 461 (37.3) | 201 (34.9) | |
| **HIV positive n (%)** | 116 (1.7) | 295 (0.6) | | 81 (14.3) | 43 (3.5) | 28 (4.8) | <0.01 |
| **All-cause mortality n (%)** | 738 (10.9) | 1974 (4.0) | <0.01 | 619 (11.0) | 178 (14.4) | 74 (12.8) | <0.01 |
| **Receiving OAT n (%)** | 6445 (95.0) | 2457 (5.0) | <0.01 | 5411 (95.1) | 1182 (95.6) | 553 (96.0) | <0.01 |

received a diagnosis for IE, 1,236 (2.26%) patients received a diagnosis for OM, and 576 (1.03%) patients were diagnosed with septic arthritis (Table 2). The 45 to 54 year old group had the highest prevalence of deep tissue infections. The female group had a higher proportion of patients with deep tissue infections (37.9%) than no deep tissue infections (34.8%). The prevalence of deep tissue infections was highest among patients living in southern urban locations and in the lowest income group. There was a higher prevalence of HIV and all-cause mortality in the deep tissue infections group.

The incidence of deep tissue infections increased over time with a particularly significant increase between 2011 and 2012 and again between 2013 and 2014 (Fig 2).

In the multivariate analysis, it was observed that being over 25 years old was correlated with having a diagnosis of one or more deep tissue infections when compared to the oldest age group (age 65 and over): age 25 to 34: adjusted Odds Ratio (OR) = 1.4 (1.3–1.6), age 35 to 44, aOR = 2.3 (2.1–2.6), age 45 to 54, aOR = 3.9 (3.5–4.3) and age 55 to 64, aOR = 8.60; 95% CI = 7.42–9.96. Similar results were observed for all three types of deep tissue infections. Additionally, being female was correlated with having a diagnosis of infective endocarditis, (aOR = 1.2 (1.2–1.3)) when compared to males. However, neither location of residence or income quintile had a significant correlation with having a diagnosis of one or more deep tissue infections. Having a positive HIV status was correlated with having a diagnosis of infective endocarditis (aOR = 1.97, 95% CI = 1.52–2.54), osteomyelitis (aOR = 4.16, 95% CI = 2.97–5.82), and septic arthritis (aOR = 4.63, 95% CI = 3.1–7.0) compared to those who had an HIV

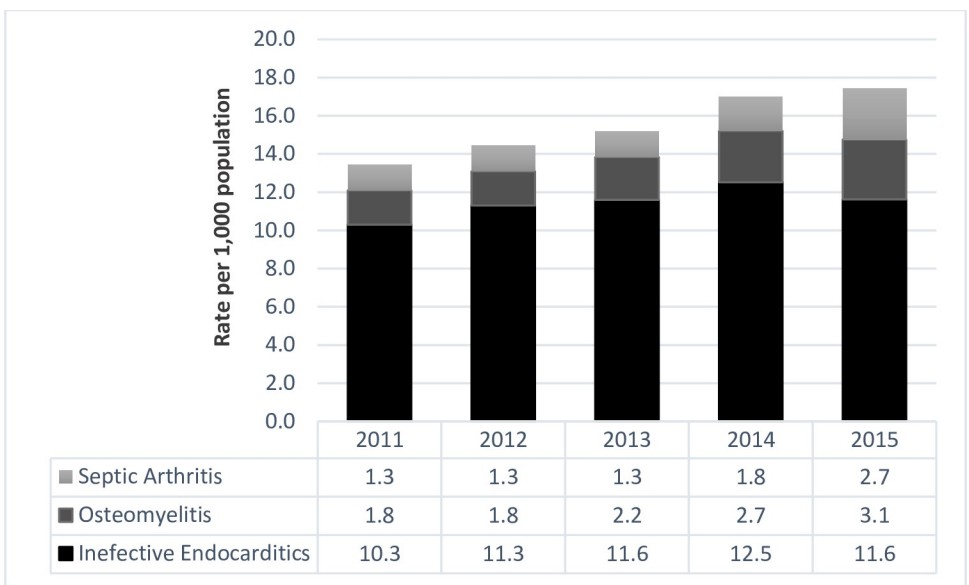

**Fig 2. Incidence rate of deep tissue infections in the OAT population in Ontario.**

negative status. All-cause mortality was correlated with a diagnosis of infective endocarditis (aOR = 1.9, 95% CI = 1.7–2.1), osteomyelitis (aOR 2.5, 95% CI = 2.1–2.9), and septic arthritis (aOR = 1.9; 95% CI = 1.4–2.4). Moreover, receiving OAT was correlated with a reduction in the likelihood of a diagnosis of infective endocarditis (aOR) = 0.7, 95% CI = 0.6–0.9), osteomyelitis (aOR = 0.9, 95% CI = 0.9–0.9) and septic arthritis (aOR = 0.9, 95% CI = 0.9–0.9). Results are presented in Table 3.

## Discussion

Our study demonstrated an increasing incidence of deep tissue infections in patients on OAT in Ontario. Additionally, the results of this study indicated that certain patient characteristics such as age, sex, concurrent infections, and all-cause mortality are correlated with having a diagnosis of one or more deep tissue infections diagnoses. The results of this study also demonstrated that being actively enrolled in OAT was correlated with a decreased likelihood of having a diagnosis of one or more deep tissue infections.

In line with our hypothesis, our analysis indicated a steady increase in the incidence of deep tissue infections with a significant rise between 2011 and 2012 and again between 2013 and 2014. Between 2006 and 2011, Oxycodone became much more available, and oxycodone deaths began to rise [75].

In 2012, the Ontario Ministry of Health and Long-Term Care withdrew OxyContin from the Ontario Drug Benefit Formulary to combat the growing problem of opioids in Ontario [76]. Despite this intervention, opioid use continued to increase. Some studies have shown that individuals shifted towards other opioids such as heroin shortly after OxyContin prescriptions became more difficult to obtain [77–79]. Particularly, in Ontario, hydromorphone, morphine and fentanyl prescriptions rose by 79% and 20% respectively between 2009 and 2013 [75, 78, 80]. These other opioids are more often injected than oxycodone when misused so we considered there might have been a switch from non-injection to injection use among people with OUD as shown in the United States under similar circumstances [81]. In this study, we focused on a population of people with OUD, in Ontario over the time when this change was

**Table 3. Correlation between patient characteristics and deep tissue infections.**

| Variable | Infective Endocarditis | | Osteomyelitis | | Septic Arthritis | |
|---|---|---|---|---|---|---|
| | OR (95% CI) | Adjusted OR (95% CI) | OR (95% CI) | Adjusted OR (95% CI) | OR (95% CI) | Adjusted OR (95% CI) |
| **Age** | | | | | | |
| 15 to 24 | 0.9 (0.8–0.9) | 0.9 (0.8–0.9) | 1.4 (1.1–1.7) | 1.4 (1.1–1.7) | 1.3 (0.9–1.8) | 1.3 (0.9–1.8) |
| 25 to 34 | 1.4 (1.3–1.6) | 1.4 (1.3–1.6) | 2.9 (2.3–3.6) | 2.8 (2.2–3.5) | 3.1 (2.2–4.3) | 2.7 (2.0–3.8) |
| 35 to 44 | 2.4 (2.2–2.7) | 2.3 (2.1–2.6) | 4.4 (3.5–5.5) | 3.9 (3.1–5.0) | 3.9 (2.8–5.4) | 3.2 (2.3–4.4) |
| 45 to 54 | 4.2 (3.8–4.7) | 3.9 (3.5–4.3) | 7.0 (5.4–9.0) | 6.0 (4.7–7.8) | 5.1 (3.5–7.4) | 4.1 (2.8–6.1) |
| 54 to 65 | 10.0 (8.7–11.6) | 8.6 (7.4–10.0) | 8.3 (5.8–11.8) | 6.6 (4.6–9.4) | 10.2 (6.4–16.1) | 8.3 (5.2–13.3) |
| 65+* | Ref | Ref | Ref | Ref | Ref | Ref |
| **Sex** | | | | | | |
| Male* | Ref | Ref | Ref | Ref | Ref | Ref |
| Female | 1.2 (1.1–1.3) | 1.2 (1.2–1.3) | 1.0 (0.9–1.2) | 1.1 (1.0–1.2) | 1.0 (0.9–1.2) | 0.8 (0.7–1.0) |
| **Place of residence** | | | | | | |
| Southern Urban* | Ref | Ref | Ref | Ref | Ref | Ref |
| Northern Rural | 0.7 (0.6–0.8) | 0.8 (0.7–0.9) | 0.9 (0.8–1.3) | 1.3 (1.0–1.7) | 0.6 (0.4–0.9) | 0.6 (0.4–0.99) |
| Northern Urban | 0.7 (0.6–0.8) | 0.8 (0.7–0.9) | 1.0 (0.8–1.2) | 1.2 (0.9–1.4) | 1.3 (1.1–1.7) | 1.3 (1.0–1.6) |
| Southern Rural | 0.9 (0.9–1.0) | 1.1 (0.9–1.2) | 0.7 (0.6–0.9) | 0.9 (0.7–1.1) | 0.8 (0.6–1.1) | 0.8 (0.6–1.1) |
| **Income** | | | | | | |
| 5 (highest)* | Ref | Ref | Ref | Ref | Ref | Ref |
| 4 | 0.9 (0.8–1.0) | 0.9 (0.8–1.0) | 1.1 (0.9–1.4) | 4.2 (3.0–5.8) | 1.0 (0.7–1.4) | 1.0 (0.7–1.4) |
| 3 | 0.9 (0.8–1.0) | 1.0 (0.9–1.1) | 1.0 (0.8–1.3) | 1.0 (0.8–1.3) | 1.1 (0.8–1.5) | 1.0 (0.7–1.4) |
| 2 | 0.9 (0.8–1.0) | 0.9 (0.9–1.1) | 1.2 (0.9–1.5) | 1.2 (0.9–1.4) | 1.3 (0.9–1.7) | 1.1 (0.8–1.6) |
| 1 (lowest) | 0.9 (0.9–1.0) | 1.0 (0.9–1.1) | 1.3 (1.1–1.6) | 1.3 (1.0–1.6) | 1.2 (0.9–1.6) | 1.0 (0.8–1.4) |
| **HIV positive** | 2.2 (1.7–2.8) | 1.8 (1.4–2.3) | 5.3 (3.9–7.3) | 4.2 (3.0–5.8) | 7.3 (4.9–10.9) | 4.6 (3.1–7.0) |
| **ACM** | 2.8 (2.6–3.1) | 1.9 (2.0–2.1) | 4.0 (3.4–4.7) | 2.5 (2.1–2.9) | 3.5 (2.8–4.5) | 1.9 (1.4–2.4) |
| **Receiving OAT** | 0.6 (0.5–0.9) | 0.7 (0.6–0.9) | 0.7 (0.6–0.8) | 0.9 (0.9–0.9) | 0.6 (0.4–0.9) | 0.9 (0.9–0.9) |

occurring to see if there was a switch from non-injection to injection misuse as reflected in an increase in the number of cases of deep tissue infections. Our findings support this increase in deep tissue infections occurred which is likely an unintended consequence of the change in the formulary. The shift to injection use may also have laid the groundwork for the introduction of injection use of illicit fentanyl as the availability of morphine and hydromorphone has declined with changes in physician prescribing practices [82].

Interestingly, being over the age of 25 and under 65, as well as being female were characteristics that were correlated with having a diagnosis of one or more deep tissue infections in this study population. The age association may be explained by the fact that age can also be associated with a longer period of drug use, and that life expectancy is lower in our population [83, 84]. Moreover, we found that being female was correlated with an increased likelihood of having a diagnosis of one or more deep tissue infections. This finding is important because it has been shown that females are more likely to be injected by their male sexual partners [29, 85], and more likely to share injection equipment [29], which places them at higher risk of infections. These finding can help inform policies and procedures to prevent infections by implementing targeted interventions when patients have health care encounters in the emergency department, hospitals, treatment facilities, or other community programs. Such harm reduction strategies can include: the provision of clean tourniquets, alcohol pads, sterile water, and needles, providing education on safe injecting practices, and having supervised consumption sites available to people who inject drugs.

The results of this study demonstrated that being actively enrolled in OAT was correlated with a decreased likelihood of having a diagnosis of one or more deep tissue infections. Other authors showed similar benefits to receiving OAT for other infections such as HIV and hepatitis C [55, 86–89]. However, there have been limited studies demonstrating the impact of OAT on the risk of deep tissue infections. Interestingly, other studies examining deep tissue infections among injection drug users call for enhanced access to addiction treatment to address the underlying causes of such infections [26, 90, 91]. Additional studies are needed to support our findings.

Our study has limitations that are important to discuss. One of the potential issues inherent to any study of this type is that health administrative data were not collected to do research. Thus the accuracy may have led to misclassification of disease prevalence and clinical outcomes. Moreover, the use of a secondary data source limited our study to the use of physician billing. Billing is dependent on the accuracy and reliability of recording practices on behalf of physicians. Additionally, details such as years of drug use, the amount or type of opioid used, history of mental health services one year before their first episode of OAT, and the number of times patients were in and out of OAT after their first episode of OAT remain unknown. There are also limitations in only having patient characteristic data and not data on drug taking and harm reduction behaviour such as mode of drug taking (injecting versus non-injecting), homelessness, access or engagement with other harm reduction (e.g. sterile needles/syringes, wipes). Since this study is observational in nature, causality cannot be inferred. Additionally, due to the large sample size, there is a potential for associations to be significant by chance. For these reasons, the results of this study which were found to be statistically significant must be interpreted critically within the context of the population of interest to determine whether the results have a clinical or health system impact.

## Conclusion

The results of this study suggest that the incidence of deep tissue infections has risen over the last five years and that patient characteristics such as age and sex are correlated with having a diagnosis of one or more deep tissue infections. Moreover, the results demonstrate that active engagement in OAT is correlated with a decreased likelihood of deep tissue infections. A better understanding of other factors that contribute to the increasing incidence of deep tissue infections and further study on the impact of OAT on deep tissue infections is needed.

## Supporting information

**S1 Appendix. Summary of study variables, type, sources and variable format for analysis.** (DOCX)

## Acknowledgments

We thank IC/ES Data Analytic Services, more specifically, Ryan Ng for his assistance with data extraction and database set up. We also thank members of the Patient and Family Advisory Committee for sharing their stories and helping to guide the research project. Lastly, we would like to thank Brittany A Loney for editing the final draft of the paper.

## Author Contributions

**Conceptualization:** Kristen A. Morin, Chad R. Prevost, Joseph K. Eibl, Michael T. Franklyn, Alexander R. Moise, David C. Marsh.

**Data curation:** Kristen A. Morin.

**Formal analysis:** Kristen A. Morin, Chad R. Prevost.

**Funding acquisition:** Kristen A. Morin, Joseph K. Eibl, Michael T. Franklyn, David C. Marsh.

**Investigation:** Kristen A. Morin, Joseph K. Eibl, Michael T. Franklyn, Alexander R. Moise, David C. Marsh.

**Methodology:** Kristen A. Morin, David C. Marsh.

**Project administration:** Kristen A. Morin, David C. Marsh.

**Resources:** Joseph K. Eibl, David C. Marsh.

**Software:** Kristen A. Morin.

**Supervision:** Joseph K. Eibl, Michael T. Franklyn, Alexander R. Moise, David C. Marsh.

**Validation:** Kristen A. Morin.

**Writing – original draft:** Kristen A. Morin, Chad R. Prevost.

**Writing – review & editing:** Kristen A. Morin, Joseph K. Eibl, Michael T. Franklyn, Alexander R. Moise, David C. Marsh.

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
