## [Decision Letter · Decision Letter 0]

3 Mar 2020

PONE-D-19-34949

A Retrospective Cohort Study Evaluating Correlates of Deep Tissue Infections Among Patients Enrolled in Opioid Agonist Treatment Using Administrative Data in Ontario, Canada

PLOS ONE

Dear Dr. Marsh, 

Thank you for submitting your manuscript to PLOS ONE. After careful consideration, we feel that it has merit but does not fully meet PLOS ONE’s publication criteria as it currently stands. Therefore, we invite you to submit a revised version of the manuscript that addresses the points raised during the review process.

We would appreciate receiving your revised manuscript by April 1, 2020.  To enhance the reproducibility of your results, we recommend that if applicable you deposit your laboratory protocols in protocols.io, where a protocol can be assigned its own identifier (DOI) such that it can be cited independently in the future. For instructions see: http://journals.plos.org/plosone/s/submission-guidelines#loc-laboratory-protocolsPlease include the following items when submitting your revised manuscript:A rebuttal letter that responds to each point raised by the academic editor and reviewer(s). This letter should be uploaded as separate file and labeled 'Response to Reviewers'.A marked-up copy of your manuscript that highlights changes made to the original version. This file should be uploaded as separate file and labeled 'Revised Manuscript with Track Changes'.An unmarked version of your revised paper without tracked changes. This file should be uploaded as separate file and labeled 'Manuscript'.

We look forward to receiving your revised manuscript.

Kind regards,

Javier Cepeda

Academic Editor

PLOS ONE

Journal Requirements:

"Dr. David Marsh maintains the following roles: Chief Medical Director at CATC (Canadian Addiction Treatment Center), opioid agonist therapy provider. Dr. Marsh has no ownership stake in the CATC as a stipendiary employee. We do not foresee any conflict of interest as data will be made freely available to the public and the CATC, and the Universities have no ability to prevent publication and dissemination of knowledge. The authors have no conflicts declared.".

 i) Please confirm that this does not alter your adherence to all PLOS ONE policies on sharing data and materials, by including the following statement: "This does not alter our adherence to  PLOS ONE policies on sharing data and materials.” (as detailed online in our guide for authors http://journals.plos.org/plosone/s/competing-interests).  If there are restrictions on sharing of data and/or materials, please state these. Please note that we cannot proceed with consideration of your article until this information has been declared.

 ii) Please include your updated Competing Interests statement in your cover letter; we will change the online submission form on your behalf.

4. Please include your tables as part of your main manuscript and remove the individual files. Please note that supplementary tables (should remain/ be uploaded) as separate "supporting information" files

5. We note you have included tables to which you do not refer in the text of your manuscript. Please ensure that you refer to Tables 1 and 2 in your text; if accepted, production will need this reference to link the reader to the Tables.

Reviewers' comments:

Reviewer's Responses to Questions

**Comments to the Author**

1. Is the manuscript technically sound, and do the data support the conclusions?

Reviewer #1: Yes

Reviewer #2: Yes

2. Has the statistical analysis been performed appropriately and rigorously? 

Reviewer #1: Yes

Reviewer #2: I Don't Know

3. Have the authors made all data underlying the findings in their manuscript fully available?

Reviewer #1: Yes

Reviewer #2: Yes

4. Is the manuscript presented in an intelligible fashion and written in standard English?

Reviewer #1: Yes

Reviewer #2: Yes

5. Review Comments to the Author

Reviewer #1: This paper presents a cohort study which utilises health administrative data to examine the incidence and prevalence of deep tissue infections among those enrolled on opioid agonist treatment in Ontario. The authors also undertake multi-variable analysis to examine the individual characteristics associated with deep tissue prevalence.

Deep tissue infections among drug users is a topic that is under-researched. It is an important topic as such infections can lead to mortality and morbidity. This paper contributes to the limited knowledge on this topic in particular by showing an increase in the prevalence of such infections.

However, there are a number of revisions that would improve the paper.

MAJOR REVISIONS

Introduction

Page 2, last paragraph – “We also hypothesize that individual patient characteristics such as age,

gender and location of residence, are correlated with DTIs.” - The authors do not justify or explain what research informs this hypothesis - what have studies said about sex/gender, age and residence and DTIs? In general, the authors need to make it more clear what the gaps in the research are concerning risk factors for DTIs and how this paper addresses this gap. Useful references not included in the paper are: (Larney, Peacock, Mathers, Hickman, & Degenhardt, 2017) and (Dahlman, Berge, Björkman, Nilsson, & Håkansson, 2018).

Results

There is no paragraph on Sample Characteristics. Also, did the authors have a breakdown of the number of injecting drug users and non-injecting drug users in the sample?

MINOR REVISIONS

A brief description of the DTIs that are examined would be useful for those readers not familiar with these infections.

Methods

Check through the use of abbreviations – sometimes an abbreviation was not given in full – e.g. page 3 line 60 – ICES; page 4 line 67 - OHIP.

Page 4 line 73 - Information on databases and study variables are available. – Where are they available.

Page 4 Cohort Definition Line 76-81: A brief definition of the codes used would be useful – maybe in a Table.

 

Results

line 195-201 – Also include the n (%) for ‘one or more DTIs’. Put in the n (%) for females (line 199), location and income groups, HIV and all cause mortality (200-201). Line 229 – spell it out how OxyContin relates to your findings

line 206 – make it clear that you are reporting the multivariable analysis – e.g. “In the multivariable analysis …”

Discussion

Line 229 – make it clear how the withdrawal of OxyContin relates to your findings

Line 258 onwards – Limitations: The authors highlight the limitations of using health administrative data for data analysis and briefly mentioned gaps in the data – they could also mention at line 263 other important variables e.g. mode of drug taking (injecting versus non-injecting), homelessness, access or engagement with other harm reduction (e.g. sterile needles/syringes, wipes). There are limitations in only having patient characteristic data and not data on drug taking and harm reduction behaviour.

Figure 1

Use footnotes to explain the abbreviations. I’m unclear about the exclusion criteria as it reads – it looks like a mix of inclusion and exclusion criteria.

Dahlman, D., Berge, J., Björkman, P., Nilsson, A. C., & Håkansson, A. (2018). Both localized and systemic bacterial infections are predicted by injection drug use: A prospective follow-up study in Swedish criminal justice clients. PLOS ONE, 13(5), e0196944. doi:10.1371/journal.pone.0196944

Larney, S., Peacock, A., Mathers, B. M., Hickman, M., & Degenhardt, L. (2017). A systematic review of injecting-related injury and disease among people who inject drugs. Drug & Alcohol Dependence, 171, 39-49. doi:10.1016/j.drugalcdep.2016.11.029

Reviewer #2: I think it needs you need to define at the outset that the study population is only those enrolled in treatment at some point. Therefore, this is really a different study than what you would typically expect. I'm assuming this population was chosen as a convenience sample? if there is another reason why those enrolled in treatment were selected that needs to be articulated at the outset.

In general, I have not really seen use of the phrase "DTI". SSTI, which is used in IDSA guidelines and throughout the literature is what I see more commonly. For your consideration.

Specific points:

Though perhaps obvious to some readers, I would recommend naming at some point the fact that OAT enrollment has been proven to decrease frequency of IDU but that many people continue to have IDU while on therapy.

In lines 27 through 28 you use mortality data to highlight the opioid use epidemic. I understands data on number of people actually using are hard to come by as compared to mortality data, but one does not exactly equal the other. Lansky PLOS ONE 2014 has a helpful accounting of US based numbers. Do comparable Canadian data exist? If so, would be helpful to cite this.

Lines 143 -Are you able to better characterize patient HIV characteristics. You talk about numbers with HIV --do you have data on who at least had a documented HIV test within a defined period (1 year, 3 years). Were there people with positive HIV tests,.Again, the methods here really appear to select for a patient population that is disparate from the most high risk for DTIs (not in OAT treatment, HIV+ or HIV status undefined, etc).

Lines 150, here as well as elsewhere you use the phrase "opioid user", which is generally not the preferred nomenclature to reference this patient population. I would recommend PWID - "persons who inject drugs", which is the phrase used by CDC, WHO, etc.

Lines 180 Recommend outlining the names of the different DTI names rather than just the codes.

Lines 247 -This sentence could use references, the phrase "giving clean needles" is an oversimplification of what is provided at syringe exchange programs, as provision of clean tourniquets, alcohol pads, sterile water etc all contribute to decreased DTI risk. Would also utilize the phrase "harm reduction" as this is really the umbrella concept you are referencing here.

6. PLOS authors have the option to publish the peer review history of their article (what does this mean?). If published, this will include your full peer review and any attached files.

Reviewer #1: No

Reviewer #2: No

---

## [Author Response · Author response to Decision Letter 0]

23 Mar 2020

Ducment provided in attached files. For formatting purposes, we did not copy and post here

---

## [Editor Report · Decision Letter 1]

31 Mar 2020

PONE-D-19-34949R1

A Retrospective Cohort Study Evaluating Correlates of Deep Tissue Infections Among Patients Enrolled in Opioid Agonist Treatment Using Administrative Data in Ontario, Canada

PLOS ONE

Dear Dr. Marsh,

Thank you for submitting your manuscript to PLOS ONE. After careful consideration, we feel that it has merit but does not fully meet PLOS ONE’s publication criteria as it currently stands. Therefore, we invite you to submit a revised version of the manuscript that addresses the points raised during the review process.

Overall:  There are too many abbreviations and acronyms used throughout the paper (e.g. OHIP, DTI, RPDB, IE, OM, SA, etc.) which could frustrate the reader.  Consider spelling out these terms instead. 

Line 75 - Authors need to provide more supporting information as to why they hypothesize that females are at higher risk of DTI.  I believe this is stated in the methods (lines 158-159). 

Line 77:  A reference is needed for this sentence. 

Line 134: If no one was excluded because there n=0 patients, then this should be deleted. 

Line 153 (and elsewhere):  "Ratio measures of effect" is not standard terminology.  Please change to "relative measure of association" throughoutLine 195:  Why were those with missing income quintile classified as the lowest income group?  Line 210:  Overdose should be one word. Lines 258 - 260.  For the logistic regression modeling, was there more than one observation per patient in the analysis?  If so, how was this accounted for?  Usually generalized estimating equations or generalized linear mixed modeling is conducted but this is not clear if these methods were used. Figure 2:  The number of cases is not particularly meaningful.  Please report the "rate" (i.e. per 1,000 population) Lines 284 - 287:  With the referent group being >65, it appears that the odds ratios increase as age increases.  Why is this? Lines 293 - 295:  Please double check all confidence intervals.  It appears that some are not correct (23.07 - 6.98). Table 3:  Please indicate the reference group with either "Ref" or "1.0".  Do not leave the row blank. Line 329:  Sentence is unclear because being over age 25 and under 65 are not mutually exclusive. Lines 332 - 336:  Sentence is too long. Please break into two to increase readability. 

We would appreciate receiving your revised manuscript by May 15 2020 11:59PM. To enhance the reproducibility of your results, we recommend that if applicable you deposit your laboratory protocols in protocols.io, where a protocol can be assigned its own identifier (DOI) such that it can be cited independently in the future. For instructions see: http://journals.plos.org/plosone/s/submission-guidelines#loc-laboratory-protocols

We look forward to receiving your revised manuscript.

Kind regards,

Javier Cepeda

Academic Editor

PLOS ONE

---

## [Author Response · Author response to Decision Letter 1]

2 Apr 2020

Thank you for your comments. Please see the "Response to Reviewers" document attatched. Response is not copied here due to formatting issues.

---

## [Editor Report · Decision Letter 2]

9 Apr 2020

A Retrospective Cohort Study Evaluating Correlates of Deep Tissue Infections Among Patients Enrolled in Opioid Agonist Treatment Using Administrative Data in Ontario, Canada

PONE-D-19-34949R2

Dear Dr. Marsh,

We are pleased to inform you that your manuscript has been judged scientifically suitable for publication and will be formally accepted for publication once it complies with all outstanding technical requirements.

With kind regards,

Javier Cepeda

Academic Editor

PLOS ONE
---

## [Editor Report · Acceptance letter]

13 Apr 2020

PONE-D-19-34949R2 

A Retrospective Cohort Study Evaluating Correlates of Deep Tissue Infections Among Patients Enrolled in Opioid Agonist Treatment Using Administrative Data in Ontario, Canada 

Dear Dr. Marsh:

I am pleased to inform you that your manuscript has been deemed suitable for publication in PLOS ONE. Congratulations! Your manuscript is now with our production department. 

With kind regards,

on behalf of

Dr. Javier Cepeda 

Academic Editor

PLOS ONE